# In Situ Measurements of Plankton Biorhythms Using Submersible Holographic Camera

**DOI:** 10.3390/s22176674

**Published:** 2022-09-03

**Authors:** Victor Dyomin, Alexandra Davydova, Nikolay Kirillov, Sergey Morgalev, Elena Naumova, Alexey Olshukov, Igor Polovtsev

**Affiliations:** 1Laboratory for Radiophysical and Optical Methods of Environmental Research, National Research Tomsk State University, 36 Lenin Avenue, 634050 Tomsk, Russia; 2Laboratory of Ichtyology, Limnological Institute SB RAS, 3 Ulan-Batorskaya Street, 664033 Irkutsk, Russia

**Keywords:** submersible holographic camera, plankton, submersible plankton sensor, plankton migration, ecosystem monitoring, biological rhythms, rhythmic processes, time series of measurements, harmonic analysis, environment factors, pollution bioindication

## Abstract

The paper presents a diagnostic complex for plankton studies using the miniDHC (digital holographic camera). Its capabilities to study the rhythmic processes in plankton ecosystems were demonstrated using the natural testing in Lake Baikal in summer. The results of in situ measurements of plankton to detect the synchronization of collective biological rhythms with medium parameters are presented and interpreted. The most significant rhythms in terms of the correlation of their parameters with medium factors are identified. The study shows that the correlation with water temperature at the mooring site has the greatest significance and reliability. The results are verified with biodiversity data obtained by the traditional mesh method. The experience and results of the study can be used for the construction of a stationary station to monitor the ecological state of the water area through the digitalization of plankton behavior.

## 1. Introduction 

There are two methods that are most often used to diagnose the state of aquatic ecosystems: testing impact on the system and further recording of its response, or monitoring the dynamics of bioindicator parameters formed by natural internal or external predictors. Bioindication implies the choice of an object that adequately represents the studied biome and allows obtaining statistically significant information. It assumes the choice of informative indicators or processes sensitive to changes in the state of biosystems. Quite a few studies [1,2,3,4,5] suggest that plankton should be used as an object of observation. Plankton is the most mobile (in terms of response speed) indicator of processes that is always present in the aquatic ecosystem. It is that trophic level which first perceives its changes [6,7,8], so the monitoring of plankton, diagnostics and determination of the causes of its changes can be particularly important for early warning of adverse environmental scenarios.

Environmental factors affect planktonic communities at different levels, from population and organism type to the molecular genetic level, and they can be used for early warning of environmental disasters. 

Rhythmic processes are of the utmost interest as informative indicators. If oscillations in the system are maintained due to the properties of the system itself, but not due to the effects of the periodic force, the system is called self-oscillating. Self-oscillating processes include oscillations in metabolic systems, periodic processes of photosynthesis, oscillations in calcium concentration in a cell, physiological oscillations and oscillations in the number of animals within a population or community [9,10]. 

Chronobiology considers biological rhythms as the states of biological processes in a body repeated at certain intervals. Biorhythms represent a fundamental process in wildlife. In addition, the vital activity of organisms occurs in the environment, the parameters of which change cyclically or simultaneously. Under these conditions, the internal rhythm of an organism (or the ensemble of organisms) is corrected by the external environment. Internal and external rhythms are thus synchronized. This ensures a balance both within the organism and between organisms and the environment. For instance, such relationships include circadian rhythms that demonstrate the stability and sensitivity of biosystems to environmental factors depending on the time of day [11]. 

Moreover, it is especially important in the context of bioindication that the biorhythms of planktonic organisms are sensitive to physical and chemical factors [12] and electromagnetic and gravitational conditions.

In this regard, the systems that monitor the biotic community seem quite relevant [13]. The achievements in electronics enabled the development of many in situ systems for zooplankton study [14,15,16], including underwater holography [17]. These systems are less invasive and can be automated in contrast to traditional mesh methods and techniques. This provides for continuous digitalization of plankton data directly in the habitat and opens up new prospects for further study [17,18]. 

## 2. Problem Statement 

The studies of biological rhythms to assess the state of the ecosystem require certain conditions, including: -Non-invasiveness (in terms of affecting the ecosystem) of obtaining information, which necessarily excludes the use of net catching methods;-Long-term (multiday) monitoring of representative plankton samples for the creation of time series with regular time intervals (requirement of further spectral analysis);-Parallel registration of plankton vital signs and changes in hydrological characteristics of the medium in order to detect their synchronization (asynchronization) due to adverse changes.

This paper considers the possibilities of the data measurement system based on a digital holographic camera that serves the station for monitoring and biorhythmology. To that end, the paper examines collective phenomena in the biotic community at the level of genetic memory of planktonic organisms supported by an external synchronizer, for example, sunlight. 

Plankton migrations can be such a display of external synchronization. Despite the fact that these congenital (daily, diurnal) rhythms are endogenous, they synchronize with external factors and can be disturbed over time if external environmental conditions change [9,11]. 

In this regard, the stability of rhythmic processes in changing the plankton population as a result of migrations may indicate the ecosystem state [19,20,21,22]. 

Pollution, along with illumination, temperature and salinity, can be considered as one of the environmental parameters affecting the rhythmic processes of the biotic community [23], and may lead to the asynchronization of rhythmic processes, which, in turn, can be used for diagnostic purposes [24]. In addition to reflecting the ecological state of the ecosystem, migration processes of the plankton play an important role in regulating the transfer of organic matter in the water area and characterizing the carbon cycle. 

This work assesses the capabilities of a submersible digital holographic camera and the digital underwater holography to study the laws of cyclic processes of coastal plankton in Lake Baikal under natural conditions by recording the time series of measurements using the stationary environmental monitoring station with fiber-optic and mobile communication lines. This study logically continues the cycle of our works, such as [25], where we performed the in situ registration of plankton concentrations within one day followed by laboratory data processing. In this work, we tested not only the technology of time series processing for bioindication purposes but were also able to analyze the equipment performance during long-term moorage. The functioning of all systems and correlation, processing and representativeness of data were thus ensured. This station can be used as a prototype of a stationary environmental monitoring station operating in a global data medium. 

## 3. Equipment and Methods

The holographic principle of information recording includes the digital registration of a hologram of the holographic camera working volume with subsequent numerical reconstruction of images of all suspended particles in the aqueous medium volume at the hologram registration stage. In this case, the images of all particles are reproduced regardless of the depth of the scene (working volume), while the recording stage does not require the physical focus of the optical system on the illustrated particle. The digital registration of a hologram, i.e., its representation in the form of a two-dimensional digital array of intensities, ensures its transmission through the communication system to the central processor. There, the holographic data are numerically processed and represented as an array of reconstructed images for further analysis. 

Holographic systems allow studying processes in situ without sampling and interfering with biocoenosis, which is their main advantage. However, such a real-time system requires a high-performance communication channel. Therefore, we use 1 GB/s equipment for a digital holographic system. 

Thus, digital underwater holography is a promising method for the in situ study of plankton [26,27,28,29,30,31] that allows obtaining information on the size, shape and position of each plankton individual, recognizing plankton particles and characterizing behavioral responses. These capabilities can be used to study the rhythmic processes of plankton in its habitat. 

The specifications of the small-size submersible digital holographic camera (miniDHC) are given in Table 1. MiniDHC is used in the study to record a digital hologram of the water volume with suspended particles (working volume in Table 1).

This camera implements an in-line hologram recording scheme, and the hologram is registered on a CMOS matrix (Figure 1). 

The descriptions of the camera and special aspects of its use are given in a number of publications [25,32,33,34]. Among these features is the possibility of changing the working volume ($V_{\mathrm{sm}}$) within the range of 0.5–0.01 L. 

In a number of articles, for example, in [34], we showed that the miniDHC makes it possible to determine various parameters of particles and the medium itself. Such parameters include: Plankton concentration;Distribution of plankton by main taxa;Average size and dispersion of sizes of individuals;Particle size distribution;Average size of individuals and size dispersion within main taxa;Particle size distribution within main taxa;Water turbidity;Suspension statistics (histogram by non-living particle size).

This study attempts to use the miniDHC to identify the rhythmic processes of the plankton community, as well as their changes, as a response to changes in the medium parameters.

A necessary attribute of a digital holographic camera is its software that ensures hologram recording and processing—DHC software. The software allows creating a virtual 3D image of the volume with studied particles and performing the data analysis in the context of any in situ testing. Further processing makes it possible to obtain from each hologram a sharp image of all particles in the measured volume at the stage of hologram registration (Figure 1b). The registered holograms were automatically processed in batches. It included the digital reconstruction of sharp images of particles using the standard inverse Fresnel integral transforms for all possible positions inside the registered volume. This transform creates a set of reconstructed images with a 0.1 mm step, which is analyzed through digital focusing. The planes where these algorithms detected the sharp images of particles were chosen from this array using standard digital algorithms of automatic focusing. All reconstructed sharp images of particles were placed on the same plane and their three-dimensional coordinates were saved in the file. A 2D image is built based on the processing results (i.e., putting all sharp images of particles into one plane), which is used to improve the quality of images, as well as to recognize and classify particles. 

The data for each particle of the 2D measured volume image are entered into the final table for the classification analysis. The classification algorithm used in this study is focused on dimensional and morphological parameters of particles [25]. These data are used to calculate the required characteristics of the plankton community depending on the measurement task. 

Works [25,33] show that plankton taxa, which are most important in terms of determining the ecosystem equilibrium, can be reliably determined using the chosen algorithm.

The ecosystem monitoring requires only a plankton DHC sensor to study the dynamics of plankton parameters and its behavioral responses. However, in order to take into account the changes associated with the hydrophysical parameters of the medium itself, we shall reinforce the DHC by adding other sensors, and assemble a hydro-physical-chemical-biological probe to study the modes and stimuli [32]. 

These modes and stimuli are typically pressure, temperature, salinity, chlorophyll content, irradiation, dissolved oxygen level and organic content, characterizing external processes in situ. 

The probe used in this study is called the DHC FOCL probe and includes a fiber-optic communication line for continuous data transmission to the processing server. The collected data are analyzed using the DHC software making it possible to obtain, structure and present the analytical information on the studied processes. In terms of plankton, this is the number of particles in the test (working) volume, with the possibility of separation into COPEPODA and CLADOCERA taxa representing the most numerous groups of zooplankton. 

Previously performed marine studies [33] allowed establishing the optimal procedure for plankton sensor sampling to detect the circadian rhythms: 24–50 samples per day. 

The water area of Lake Baikal near the experimental station of the Limnological Institute (LIN SB RAN) in Bolshiye Koty settlement was chosen for natural testing of the miniDHC and the DHC software capabilities to study the rhythmic processes of plankton (Figure 2). 

For the experiment, the equipment was installed at the bottom station at an altitude of 2.5 m from the bottom, and the depth of measuring equipment was 2.5 m. In addition to the miniDHC, the following sensors measuring the medium parameters were connected to the DHC FOCL probe: depth pressure transducer, temperature sensor, water conductivity sensor, day/night sensor. Such configuration of the sensor assembly was chosen to study the environmental parameters through plankton measurements.

Most modern measuring devices (temperature, pressure, conductivity, illumination, etc.) are connected to the computer using a standard serial cable connection RS232 or RS485. The disadvantages of RS232 are that it requires a point-to-point cable connection, which is limited by the cable distance (up to 15 m for RS232). In addition, each such device requires a separate cable connected to the unique RS232 port on the PC. The RS485 connection partially solves these problems. We chose a more universal solution and combined all devices through serial interface converters. They are designed to connect autonomous devices with a final RS232/RS485 interface to the local network. Then, this local network is connected through a fiber-optic communication line (FOCL), formed by fiber-optic converters and a fiber-optic cable, to the local network of the research station onshore. The length of the optical cable and the data transmission speed depend on the equipment used. 

Figure 3 shows the general layout of the DHC FOCL. The features of the station include its boundary location relative to various biotopes: coastal and open subsurface Lake Baikal. This arrangement has its advantages and disadvantages: on the one hand, it allows observing the dynamics of plankton of various biomes at the biotope boundary, and on the other hand, strong turbulent flows of lake water and thermal gradients contribute to the creation of the granular noise on images reconstructed from digital holograms—speckles. This noise is created by multiple interferences of waves scattered on the phase inhomogeneities of small turbulent water vortices. The automatic classification algorithm used in the miniDHC may falsely recognize speckles as plankton. The granules of this noise appear in the reconstructed images and lead to false activation of the recognition system. The effect of this noise on the result is statistical in nature and can be reduced through ensemble averaging. Unfortunately, it is not possible through the full-scale experiment technique, when both the object and the noise are dynamic. To minimize (reduce phase change) this location factor, the working (measured) volume of the miniDHC was chosen to be very small, $V_{\mathrm{sm}}=20\;\mathrm{mL}$ [32]. 

Data on the plankton number in the measured volume of the miniDHC were formed with a frequency of 1 h in real time on a desktop computer with remote data unloading via the Internet. The measured volume for one measurement count was increased by processing 5 sequentially recorded holograms with an interval of 5 s between adjacent holograms. Thus, taking into account this averaging, the measured volume is determined by the following relation: (1)$$V_{m}=k\cdot V_{\mathrm{sm}},$$
where k is the number of holograms used to form a single count. 

Thus, for k = 5, $V_{\mathrm{sm}}=20\;\mathrm{mL},$ the measured volume was $V_{m}=0.1\;L$. 

The registered digital holograms were processed in accordance with the scheme shown in Figure 1b using a PC with NVIDIA GPU located onshore. Digital hologram recorded in Lake Baikal and images of particles reconstructed from it is shown in Figure 4. The continuous power supply mode and data transmission via FOCL to the desktop computer ensured a continuous sequence of measurements during long moorage from 1.08.21 to 30.08.21. The processing results made it possible to form the time series of monitoring data intended for transmission over the mobile 3G-4G Internet. This scheme allows reducing the load on the communication channel with the remote center and selecting only meaningful data, which does not require a lot of effort and resources. It should be noted that this series of measurements was formed in a data processing center located 5000 km from the mooring station and the desktop computer.

The averaging of data on k registered volumes may be used to estimate the error of determining the averaged concentration data as the mean ± std. err. of mean. Let us give one result: 10,000 ± 2930 Std./m^3^ at the level of significance *p* = 0.1. Thus, the error of concentration measurements under the conditions of data averaging for 5 holograms is 30%.

Data on the daily catch of mesoplankton were additionally obtained to clarify the composition of plankton at the mooring station in Lake Baikal. The large Juday net with a diameter of 37.5 cm and a mesh size of 88 µm was used for sampling. The catching was carried out totally from the boat (bottom-surface) with 2 times repetition, thus ensuring greater representativeness over a depth of 5 m in the vicinity of the DHC location. Catching frequency was every hour. Fixed plankton samples were processed in the laboratory under a microscope. Based on his experience, the expert assigned the detected organisms to any particular taxon. The results of sample processing are shown in the Figure 5. 

The basis of zooplankton was formed by various age stages of the endemic crustacean *Epischura baikalensis*, Sars G.O., 1900 belonging to the Calanoida order. 

Figure 5 suggests that changes in plankton number are mainly caused by the occurrence of epischura (*E. baikalensis*) in the coastal biotope. The rest of the plankton community near the mooring station change their number less critically; therefore, all measurements presented in the monitoring series in the next section can be attributed only to epischura. 

Epischura is a typical inhabitant of open Lake Baikal and is rarely found in the depths of isolated bays and gulfs. However, such features are also possible at the biotope boundaries, which led to the choice of the DHC FOCL probe location for the current studies. 

By its definition, epischura belongs to the COPEPODA taxon, which is reliably defined by the DHC software probe [34]. 

This belonging is visually confirmed by the photo (Figure 6) of reconstructed images from holograms registered in the water area near the mooring station.

## 4. Results and Discussion 

The following is a time sequence of measurements and its regularization. All results described in this section are based on a monitoring series of measurements of plankton $C_{\mathrm{sm}}^{0.02}$ in the measured volume $V_{\mathrm{sm}}=20\;\mathrm{mL}$ averaged over five (k = 5) sequentially recorded holograms with an interval of 1 s. At the same time, the average number is calculated using the formula: $$C_{\mathrm{sm}}^{0.02}=\frac{\sum_{i=1}^{k}C_{{\mathrm{sm}}_{i}}^{0.02}}{k}$$
where $C_{{\mathrm{sm}}_{i}}^{0.02}$ is the number of zooplankton individuals determined from the i-hologram. 

The average number is recorded every hour and displayed as a time profile (Figure 7).

This time series of measurements transmitted to the processing center does not imply classification due to the limited biodiversity of zooplankton represented mostly by only one species, *E. baikalensis*, of various development stages (Figure 5). The influence of other inhabitants and other particles suspended in water can be eliminated by regularization—subtraction of the regression line (Figure 7b). 

Indeed, in the context of the present study the specificity of the analyzed community gives the right to regularize. It should be noted that in fact this replaces the need for species recognition. This position is true only in this case and is not intended to be universal, but greatly saves computing resources and allows somewhat smoothing of the errors caused by the effect of the measured volume smallness on the final result. 

Figure 8 shows the comparison of net and holographic studies for the same day—3.08.21. Here, we took 1 L of the medium volume for data comparison. The DHC data look transformed upwards, but almost completely repeat the dynamics of net methods. This is very important for the correlation analysis of the medium significant factors. The amplitude of oscillations in plankton number in 1 L over the 23 h of the circadian rhythm is indicated as $A_{\mathrm{mv}},$ is approximately equal to the maximum amount of plankton and can be, in accordance with the Figure 3, taken as the value of the diurnal migration of the epischura, $A_{\mathrm{mv}}\approx A_{N}$. 

Rhythms in the time series of measurements. The Fourier analysis (harmonic analysis, FFT) is used and is best applied in cases of a periodically repeated process in order to distinguish significant regular components [35] against the background of random elements. 

In accordance with the above, a regularized series of measurements from Figure 7b was processed. The processing results are shown in Figure 9.

The Fourier analysis shown in Figure 9 indicates that the rhythmic processes are clearly observed in the plankton number change, of which the six most significant are distinguished by the level of statistical significance at *p* = 0.05. Their positions in the spectrum are marked red in Figure 9 and correspond to the known rhythms of living organisms described in numerous works on chronobiology [20,36,37,38,39]. 

The strongest of them is circadian rhythm with a period of 23 h. It is known that circadian rhythms rarely correspond exactly to 24 h, but most often range from 23 to 25 h [20,39,40,41]. 

Note that the circatidal (from Eng. *tidal*) rhythm lasts 12.4 h. By definition, it is associated with tides and is independent of the circadian rhythm. Apparently, this interpretation may also include the ultradian rhythm that lasts 10 h [19]. 

Ultradian rhythms with a period of 7.7 and 4.4 h are also interesting for ecosystem analysis due to the high frequency and the possibility of using the Fourier analysis to determine the average daily and average night amplitude of rhythms. 

Infradian rhythmic processes in this experiment are also partially recorded, despite the regularization and insufficient duration of the time series of measurements to detect them. This is not sufficient for interpretation in accordance with the sampling theorem but does not mean that low-frequency infradian rhythms are not informative. Thus, we need to increase the duration of the time series of measurements, for example, ensuring continuous monitoring throughout the experiment. 

In addition, earlier [33] we showed that the measured volume should be increased to 10 L for data representativeness, which means that this measuring configuration requires further processing of at least 500 holograms per hour. 

Circadian (diurnal) rhythms are typical for almost all organisms, including single-celled akaryocytes, and they are quite well studied. For this reason, we chose them to demonstrate the capabilities of the DHC diagnostics. In addition, the measured parameter is the amplitude of changes in the number of *E. baikalensis* in 1 L, $A_{\mathrm{mv}}$, determined as the difference between the maximum and minimum measurements per day, as shown in Figure 8. 

The parameters of the circadian rhythm and its natural background presented as daily graphs are shown in the Figure 10. 

The correlation of the cycle amplitude with medium parameters was analyzed by the correlation analysis using Statistica 10 standard software (StatSoft Inc., Tulsa, OK, USA). The results of the analysis are presented in Table 2. 

The high value (−0.74) of the correlation function of the amplitude of the *E. baikalensis* circadian rhythm with water temperature indicates that this is the most significant of all medium factors. Therefore, it is this correlation relationship that shall be used for pollution bioindication and ecosystem diagnostics. This is especially true in connection with the prompt response of the community to this factor. The response time (time gap between medium change and plankton change) is 2 days. 

It should be noted that the negative relationship of planktonic biome concentration with temperature and conductivity is described in many works, including the global work [44]. 

Data on Fourier processing of the time function of the background value of plankton number by days are given in Table 3. The analysis of the table data suggests that there are several rhythmic processes in changing the plankton number. *E. baikalensis* is not involved in these processes. Since the structure of the coastal plankton community is quite diverse, the rhythmic processes of other taxa have other features, in particular diurnal ultradian rhythms. In this case, the studied rhythm is formally represented by the harmonics of the daily time function of plankton concentration. 

The Fourier analysis makes it possible to determine the characteristics of these harmonics: amplitude, frequency, amplitude and phase relative to the origin [35]. As an example, Table 3 shows these characteristics for several days of moorage and highlights the most significant diurnal components. Statistica 10 (StatSoft Ink., USA) was also used for this processing.

The most reliable is the ultradian rhythm with a period of 4 h in Figure 9. It is quite pronounced, i.e., it has the proper level of significance at *p* ≤ 0.05 and can also be used as the significant parameter for diagnostic tasks. The parameters (amplitude, phase and frequency) of the fourth cycle with a period of about 4 h in the form of graphs by all days of moorage are shown in Figure 11. 

The analysis of the effect of the medium factors on the amplitude of the ultradian cycle (4 h) is presented in Table 4. The phase and frequency of the ultradian rhythm (4 h) were in no way related to the parameters of the medium, therefore, the results of the correlation analysis are not given here. 

The amplitude of the ultradian cycle (4 h) best correlates with water temperature, water conductivity, wind direction and water level in Lake Baikal. Moreover, the last two correlations are quite obvious, because in Lake Baikal it is the wind, i.e., its direction, which changes the temperature and water inflow of the inflowing rivers [45]. However, the most significant factor is the temperature. 

Note that the optimal time shift in calculating the cross-correlation function has the same value, 2 days for both analyzed rhythms, despite the fact that they belong to different dwellers. Plankton process conductivity (salinity) much longer. Changes appear within 5 days. It is also important that the values of the maxima of the cross-correlation function for water temperature coincide and are equal to 0.74. 

These studies make it possible to conclude that the correlation between both the circadian and ultradian cycles (4 h) with temperature was the most significant in the monitoring of rhythmic processes of the coastal plankton of Lake Baikal in the mooring station of a digital submersible holographic camera. Hence, in a given water area the ecological diagnostics device must determine the value of one of these correlation coefficients. Phase and frequency are not quite informative enough for the migration process. 

## 5. Conclusions

This paper presents the results of the natural testing of the stationary ecological monitoring station (DHC FOCL probe) using fiber-optic and mobile communication systems and a submersible digital holographic camera (miniDHC) operating in a stationary position for a long time. 

The construction of a long-term time series of measurements in the DHC monitoring mode allows detecting and monitoring collective synchronization of plankton biorhythms. It can be considered as an indicator of the ecological state of the water area. 

The Fourier analysis of the time series of the DHC measurements of plankton number clearly identifies its main rhythms. The most significant are biological rhythms lasting ~23 h and ~4 h, known in chronobiology as the circadian and ultradian rhythms of living organisms. The amplitude of these rhythms has the most meaningful correlation with water temperature. In these circumstances, the plankton response falls behind the temperature change by 2 days. 

The dominant role in the study of circadian rhythms belongs to *E. Baikalensis*. Further studies of the correlation of these rhythms with medium parameters require a longer series of the DHC measurements with an increase in the measured volume of the DHC. 

## Figures and Tables

**Figure 1 sensors-22-06674-f001:**
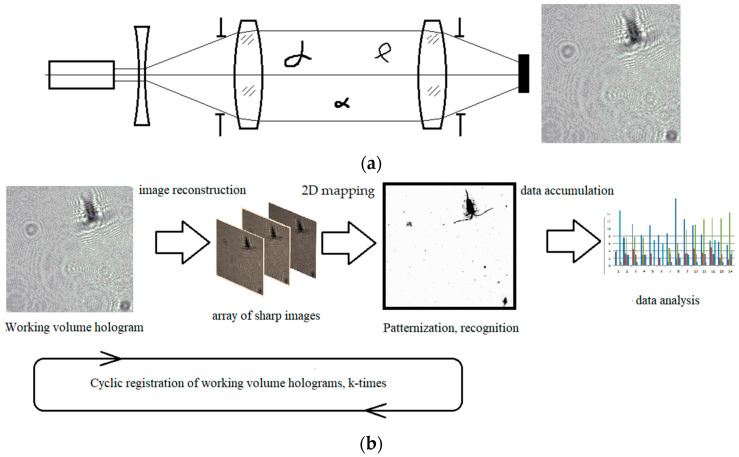
(**a**)—Hologram recording by the miniDHC; (**b**)—data extraction from a hologram for the analysis of plankton characteristics.

**Figure 2 sensors-22-06674-f002:**
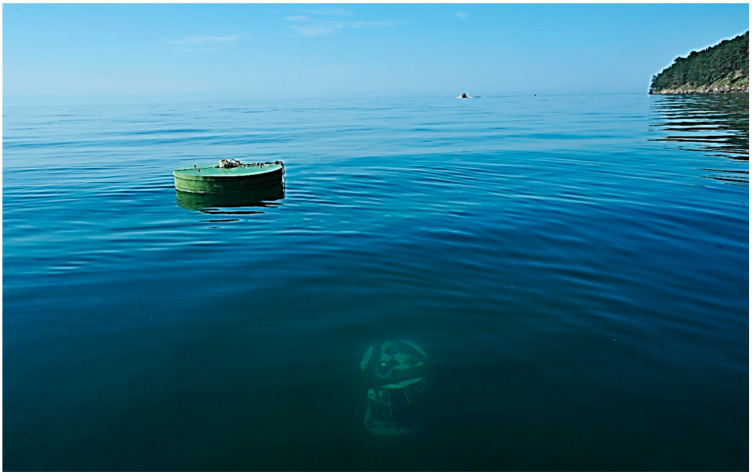
DHC FOCL probe station at the point with coordinates 51°53,952′ N 105°03,834′ E, Lake Baikal.

**Figure 3 sensors-22-06674-f003:**
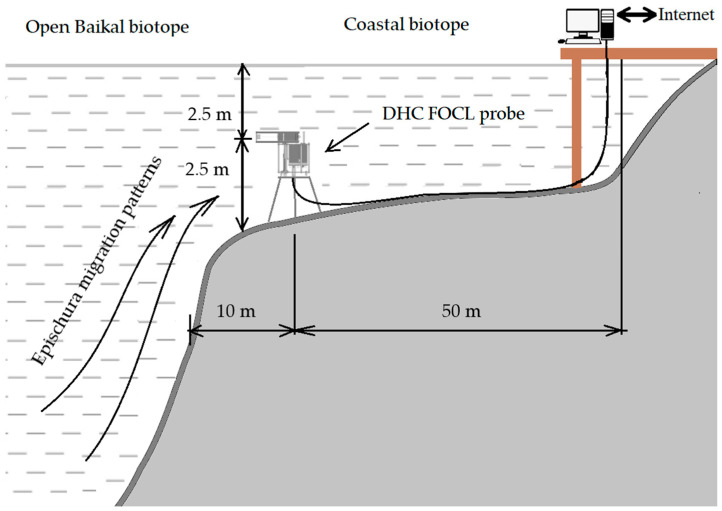
General layout and installation diagram of the DHC FOCL probe in Lake Baikal, Bolshiye Koty settlement, LIN SB RAS research station at the point with coordinates 51°53,952′ N 105°03,834′ E.

**Figure 4 sensors-22-06674-f004:**
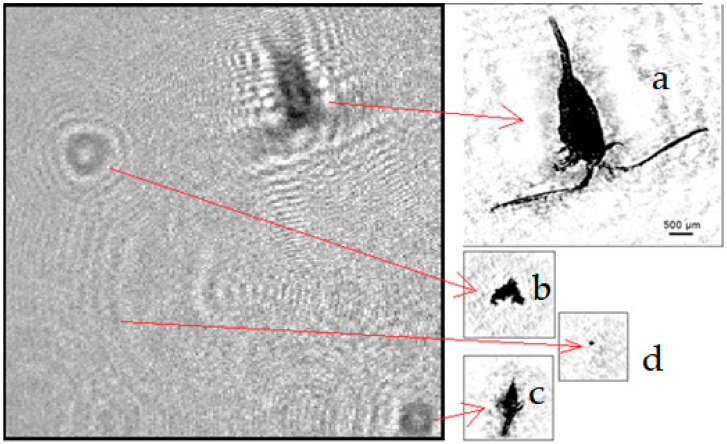
Digital hologram recorded in Lake Baikal and images of particles reconstructed from it: a—epischura, c—epischura nauplius, b and d—suspension particles.

**Figure 5 sensors-22-06674-f005:**
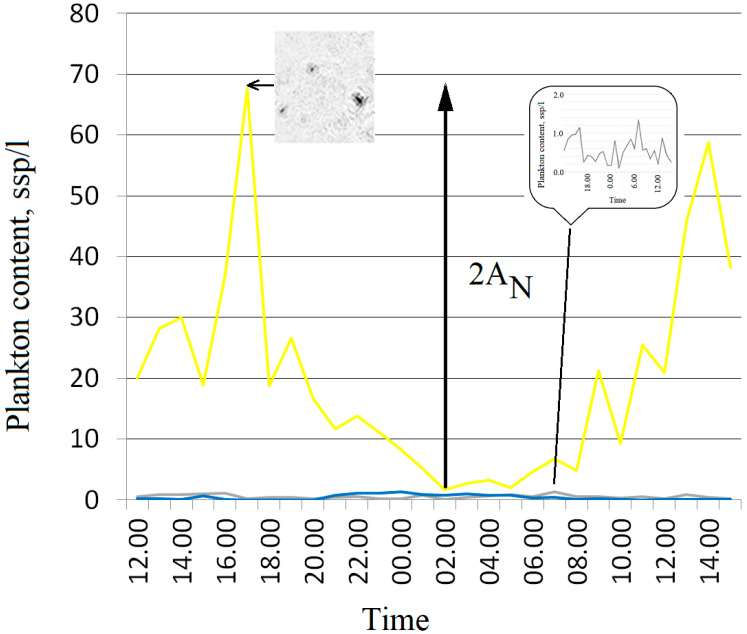
Daily changes in the composition of coastal plankton of Lake Baikal according to the net technology data. Yellow line—epischura (COPEPODA taxon), mainly of the naupliar stage; gray—rotifers (ROTIFERA taxon); blue—bottom copepods, Harpacticoida order in a water column (5-0 m). $A_{N}$—amplitude of epischura abundance fluctuations in 1 L of water per day, 3 September 2021.

**Figure 6 sensors-22-06674-f006:**
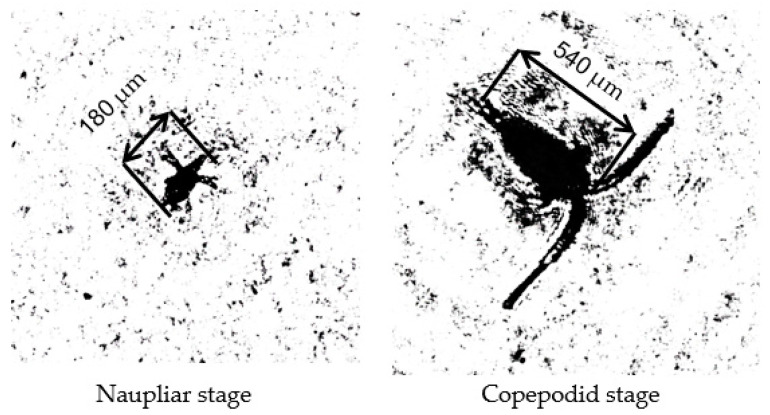
Holographic images of Baikal epischura of different ages obtained using in situ DHC technology during testing in Lake Baikal.

**Figure 7 sensors-22-06674-f007:**
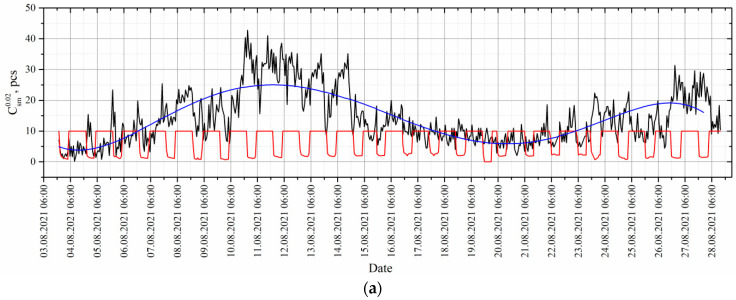
(**a**)—time series of measurements formed by the DHC FOCL probe during moorage in Lake Baikal at Bolshiye Koty Scientific Research Station (Limnological Institute (LIN)) on 3–28 August 2021. Red line—operation of the day/night sensor. Blue line—regression line. (**b**)—regularized series of measurements. $C_{\mathrm{sm}}^{0.02}$.

**Figure 8 sensors-22-06674-f008:**
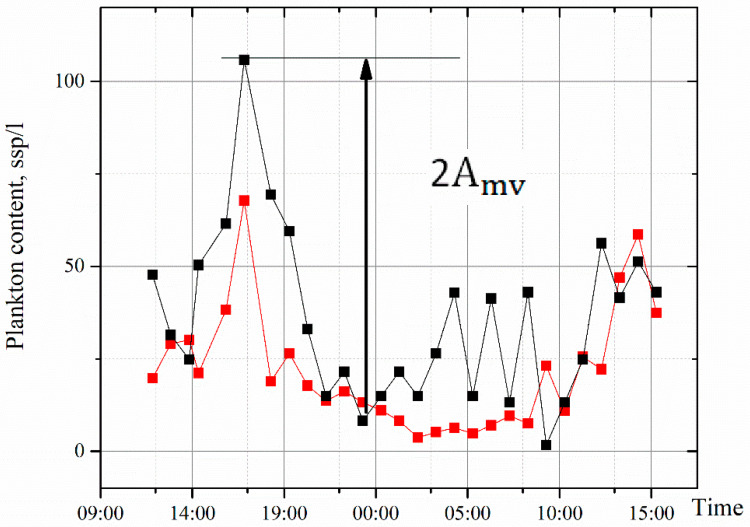
Plankton content in 1 L near the mooring station. Red line—result after processing the samples obtained by a net. Black line—time series of holographic measurements of plankton number over the same period of time.

**Figure 9 sensors-22-06674-f009:**
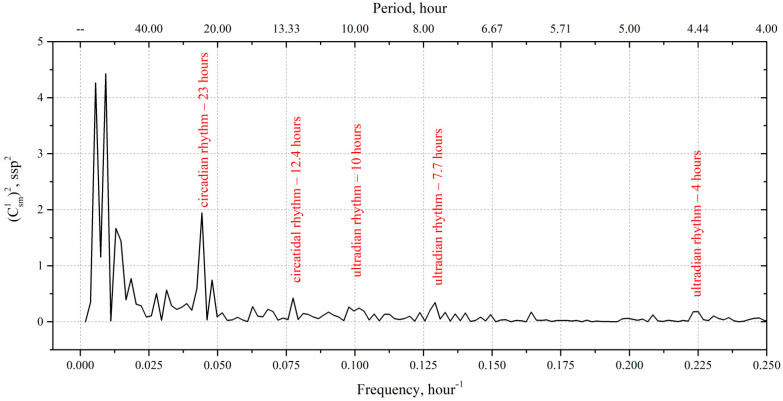
Spectrum structure of the time series of measurements. Black line—spectral density of the plankton number shown in Figure 7b, recalculated per 1 L of water.

**Figure 10 sensors-22-06674-f010:**
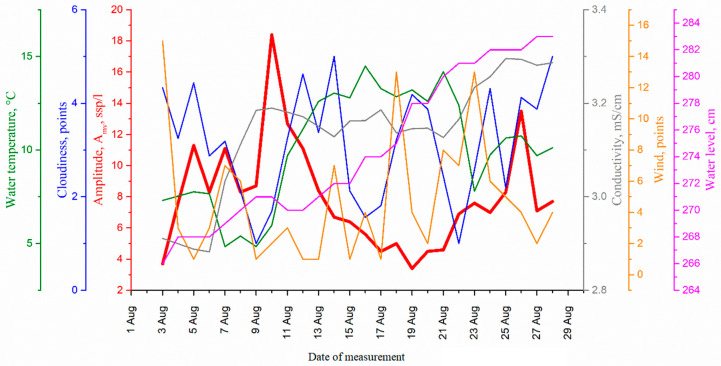
Parameters of the plankton circadian rhythm 3 August 2021 to 27 August 2021 against the background of habitat characteristics: wind direction, cloudiness, water temperature and water salinity of Baikal Lake (Bolshiye Koty mooring station). Weather data (wind direction, cloudiness) adapted with permission from Ref. [42] 2004–2022, Raspisaniye Pogodi Ltd. Water level adapted with permission from Ref. [43] 2015–2022, AllRivers.info. Water temperature and conductivity were taken from the DHC FOCL probe sensors.

**Figure 11 sensors-22-06674-f011:**
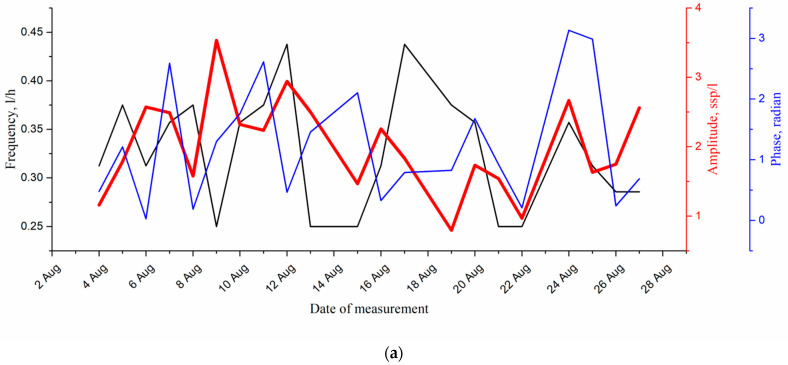
(**a**)—parameters of the ultradian rhythm (4 h) of coastal plankton over a period from 3 August 2021 to 27 August 2021; (**b**)—habitat characteristics: wind direction, cloudiness, water temperature, water conductivity. Bolshiye Koty mooring station.

**Table 1 sensors-22-06674-t001:** MiniDHC specifications.

Parameter	Value
Power characteristics: —Power voltage, V —Power consumption, W	12 20
Maximum volume studied (working volume) per one exposure, l	0.5
Provided averaging volume, l	5
Maximum working volume length, mm	338.4
Submersion depth, m, not more than	500
Size of measured particles, mm	0.1–28
Submersion speed during vertical probing, m/s	0.1–1.0
Discreteness of counts formed in real time at the submersion speed of 0.3 m/s, m	6
Ethernet transmission rate, GB/s	1
Overall dimensions (length × diameter), mm	320.5 × 142
Weight, kg, not more than	9

**Table 2 sensors-22-06674-t002:** Correlation analysis of the amplitude of *E. baikalensis* circadian rhythm with medium parameters. Red color shows the significant values of correlation coefficients, the level of significance of which is *p* ≤ 0.05, and which are related to the corresponding medium parameters.

Medium Parameter	Correlation Coefficient for Circadian Rhythm Amplitude with Medium Parameters	Optimal Time Shift of the Signal Reflecting the Change in the Medium Parameter, Day	Cross-Correlation Function Maximum at Optimal Time Shift
Cloudiness	0.0043	−11 No signs of causal relationship	−0.383344
Water temperature	−0.62	2	−0.74
Water conductivity	0.036	5	−0.53
Wind direction	−0.29	8	0.41
Water level	−0.29	−5 No signs of causal relationship	−0.42

**Table 3 sensors-22-06674-t003:** Results of the Fourier analysis of daily time functions over 6 days of moorage. Red color shows the diurnal components of the spectrum and their characteristics with the level of significance *p* ≤ 0.05.

Date	Diurnal Period of Vertical Migrations, Hour	Frequency of Diurnal Harmonics of Migrations, Hour ^−1^	Amplitude of Diurnal Harmonics, spp
4 August 2021	8.0	0.125	1.34
	3.2	0.31	1.16
	2.0	0.5	1.94
5 August 2021	4.0	0.25	2.35
	2.67	0.375	1.79
	2.0	0.5	1.56
6 August 2021	5.3	0.188	1.69
	3.2	0.31	2.57
	2.29	0.44	1.18
7 August 2021	4.67	0.21	2.65
	2.8	0.36	2.49
	2.0	0.5	1.9
8 August 2021	5.3	0.19	1.43
	2.67	0.38	1.58
	2.0	0.5	1.2
9 August 2021	8.0	0.125	2.92
	4.0	0.25	3.53
	2.67	0.38	1.27

**Table 4 sensors-22-06674-t004:** Results of the correlation analysis of the diurnal cycle amplitude (4 h) with medium parameters. Red color indicates reliable data, yellow indicates close to reliable (*p* ≤ 0.1).

Medium Parameter	Time Shift of the Signal Reflecting the Change in the Medium Parameter, Day	Cross-Correlation Function for Medium Parameter Shift
Cloudiness	0	−0.05 ± 0.2
	1	−0.30 ± 0.2
Water temperature	0	−0.58 ± 0.2
	2	−0.74 ± 0.21
Water conductivity	0	−0.02 ± 0.2
	5	−0.53 ± 0.22
Wind direction	0	−0.34 ± 0.2
	7	0.40 ± 0.23
Water level	0	−0.21 ± 0.2

## Data Availability

The data presented in this study are openly available in FigShare at https://doi.org/10.6084/m9.figshare.20677689. (accessed on 31 August 2022) [46].

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
