# Peer review of "In Situ Measurements of Plankton Biorhythms Using Submersible Holographic Camera"

_sensors, 2022, doi:10.3390/s22176674_

Round 1

Reviewer 1 Report (New Reviewer)

This article reports a study of the plankton content in sea water over days, monitored by holography.

This article is the natural continuation of a series of articles from the same authors. Here, the authors focus on the dynamics of plankton content oscillations, measure their frequencies and extract correlations with environmental parameters. This setup looks efficient, and it is none-invasive. I may be useful for the community.

My only recommendation is to better introduce the holographic method. The authors refer to some of their previous publications, but they are hardly accessible. Could the authors refer to a more popular review on this digitial holographic camera? It obviously differs from what is currently admitted to be digitial holography, where a reference beam is involved to create interferences.

Also, the authors mention, at the biginning of the article, an array of images. At first glance, it wass not clear why there was an array, and which parameter was varied from one image to another. There are some phase imaging techniques that vary the focus for instance. later, on, we guess that this series of images is actually simply a movie. They authors should clarify it.

Author Response

Dear Reviewer,

We are grateful for the comments made. We hope that as a result of our efforts to adjust the manuscript to the required standards, our work has become better and will be positively evaluated by the Reviewers.

We took all comments into account. All changes in the text of the article are made in accordance with the comments and feedback of the Reviewers and marked in red in the manuscript.

Reviewers’ comments and our answers:

Point 1: This article reports a study of the plankton content in sea water over days, monitored by holography.

This article is the natural continuation of a series of articles from the same authors. Here, the authors focus on the dynamics of plankton content oscillations, measure their frequencies and extract correlations with environmental parameters. This setup looks efficient, and it is none-invasive. I may be useful for the community.

Response 1: Thanks for your kind comment.

Point 2: My only recommendation is to better introduce the holographic method. The authors refer to some of their previous publications, but they are hardly accessible. Could the authors refer to a more popular review on this digitial holographic camera? It obviously differs from what is currently admitted to be digitial holography, where a reference beam is involved to create interferences.

Response 2: It seems that our articles published in MDPI journals are quite easy to understand by the readers due to the thorough work of reviewers and editors of these journals. We refer to those articles in the text of the manuscript:

  1. Dyomin V.V. et al. Monitoring of Plankton Spatial and Temporal Characteristics With the Use of a Submersible Digital Holographic Camera // Front. Mar. Sci. 2020. V. 7. № 653. P. 1–9.

Can be found at https://www.frontiersin.org/articles/10.3389/fmars.2020.00653/full

  1. Dyomin V. et al. Underwater Holographic Sensor for Plankton Studies In Situ including Accompanying Measurements // Sensors. 2021. V. 21. № 4863. P. 1-19

Can be found at https://www.mdpi.com/1424-8220/21/14/4863/htm

We can add additional links, but then there will be too much self-citation, which contradicts the journal policy.

Point 3: Also, the authors mention, at the beginning of the article, an array of images. At first glance, it was not clear why there was an array, and which parameter was varied from one image to another. There are some phase imaging techniques that vary the focus for instance. Later, on, we guess that this series of images is actually simply a movie. The authors should clarify it.

Response 3: Due to the fact that this article focuses not on the sensor itself, but on its use, we did not consider the issues of hologram reconstruction, since this will load the main idea with too much detail. The algorithm of the holographic unit is briefly described in Figure 1. Here, an image array represents an array of reconstructed images focused in different planes. A 2D display of the volume image of the measured volume is formed from them for subsequent classification and recognition.

We clarified this issue in lines 163-164.

Reviewer 2 Report (New Reviewer)

­­­The paper reports a demonstrative application of a holographic camera for plankton biorhythms study. DH camera is identified as suitable equipment due to its features of being nondestructive and ability of after-acquisition focusing.

The paper is more like a summary report with only limited technical content.

-      Table 1 should be moved to line 128, right after it is first mentioned.

-      There is an inconsistency of “1 Gb/s” in line 118 and I “GB/s” in table 1.

-      How the additional sensors are integrated (via RS232) with the fiber optical communication line is unclear and needs more elaboration.  

-      It is expected to propose some solution to the problem of false identification of plankton from speckles induced from the turbulent water other than imposing a limitation on the measurement volume.

Author Response

Dear Reviewer,

We are grateful for the comments made. We hope that as a result of our efforts to adjust the manuscript to the required standards, our work has become better and will be positively evaluated by the Reviewers.

We took all comments into account. All changes in the text of the article are made in accordance with the comments and feedback of the Reviewers and marked in red in the manuscript.

Reviewers’ comments and our answers:

Point 1:  The paper reports a demonstrative application of a holographic camera for plankton biorhythms study. DH camera is identified as suitable equipment due to its features of being nondestructive and ability of after-acquisition focusing.

The paper is more like a summary report with only limited technical content.

Response 1: Thanks for your comment

Point 1: - Table 1 should be moved to line 128, right after it is first mentioned.

Response 1: We changed the position of Table 1 and its description.

Point 2: - There is an inconsistency of “1 Gb/s” in line 118 and I “GB/s” in table 1.

Response 2: Thanks for the comment. We fixed the misprint

Point 3: - How the additional sensors are integrated (via RS232) with the fiber optical communication line is unclear and needs more elaboration. 

Response 3: We added some details in lines 209-220 according to the comments.

Point 4: - It is expected to propose some solution to the problem of false identification of plankton from speckles induced from the turbulent water other than imposing a limitation on the measurement volume.

Response 4: Here we limited ourselves to certain restrictions on the volume of measurements. Besides, we clarified this issue in lines 235-237.

We are grateful for all the comments. We hope that as a result our work has become much better.

This manuscript is a resubmission of an earlier submission. The following is a list of the peer review reports and author responses from that submission.

Round 1

Reviewer 1 Report

The article reports on a diagnostic complex for the study of autochthonous plankton based on the miniDHC digital holographic camera, to study the rhythmic processes in plankton ecosystems.
The objective and approach are very interesting and promising, but need to be better presented.
In particular, the manuscript must be clearly distinguishable from the article "Underwater Holographic Sensor for Plankton Studies In Situ including Accompanying Measurements" already published in 2021.

The following points need to be improved / made clearer:

(1) The introduction needs to be much better at introducing the research area and highlighting the specifics, necessity and challenges of the same. e.g., by referring to chronobiology and specific environmental factors.

(2) What are the special features (advantages, possibly also challenges) using in situ DHC technology for testing? This should be emphasized.

(3) The conclusion should also be much more informative and also address the key findings - for example, the different plankton rhythms in the context of habitat characteristics.

(4) Abstract and keywords could be significantly improved by increasing specificity.

(5) Could the key research findings be better stated in the title of the manuscript? A suggestion: "Underwater Holographic Sensor to stuy autochthonous plankton biorhythms for ecosystem diagnostics"

Author Response

Response to Reviewer 1 Comments

The article reports on a diagnostic complex for the study of autochthonous plankton based on the miniDHC digital holographic camera, to study the rhythmic processes in plankton ecosystems.
The objective and approach are very interesting and promising, but need to be better presented.
In particular, the manuscript must be clearly distinguishable from the article "Underwater Holographic Sensor for Plankton Studies In Situ including Accompanying Measurements" already published in 2021.

The following points need to be improved / made clearer:

We tried to do this in the text of the article and reflect it in our responses to the comments of the Reviewers, as well as in the edited title of the article: “In situ Measurements of Plankton Biorhythms using Submersible Holographic Camera”. The present article describes the use of a sensor based on a digital holographic camera for measuring plankton biorhythms. This topic has not been studied in the above-mentioned article.

Point 1.  The introduction needs to be much better at introducing the research area and highlighting the specifics, necessity and challenges of the same. e.g., by referring to chronobiology and specific environmental factors.
We revised the text of the introduction (lines 28-66) in accordance with the Reviewer’s comments.

Point 2.  What are the special features (advantages, possibly also challenges) using in situ DHC technology for testing? This should be emphasized.

We added new material on lines 101-114 in accordance with the Reviewer’s comments.

Point 3.  The conclusion should also be much more informative and also address the key findings - for example, the different plankton rhythms in the context of habitat characteristics.

We added new material on lines 409-415 in accordance with the Reviewer’s comments.

Point 4.  Abstract and keywords could be significantly improved by increasing specificity.

We added new material on lines 13-25 in accordance with the Reviewer’s comments.

Point 5.  Could the key research findings be better stated in the title of the manuscript? A suggestion: "Underwater Holographic Sensor to stuy autochthonous plankton biorhythms for ecosystem diagnostics"

Thank you very much for your suggestion. We made changes to the title of the article. A new edited title is as follows: “In situ Measurements of Plankton Biorhythms using Submersible Holographic Camera”.

We are grateful for all the comments. We hope that as a result our work has become much better.

Reviewer 2 Report

The paper presents a diagnostic complex for the study of autochthonous plankton based on the miniDHC digital holographic camera. It has certain novelty and significance. This paper is clearly written and well organized. The reviewer thinks this paper can be accepted for publication after some minor revisions.

(1)The optical setup diagram of the digital holographic camera may be shown, if possible.

(2)What is the advantage of digital holographic camera over other imaging or sensor devices, for this task?

(3)Some examples of recorded holograms and reconstructed images may be shown.

(4)The algorithm for estimating the plankton number from the holographic camera results is not clear.  

Author Response

Response to Reviewer 2 Comments

Comments and Suggestions for Authors

The paper presents a diagnostic complex for the study of autochthonous plankton based on the miniDHC digital holographic camera. It has certain novelty and significance. This paper is clearly written and well organized. The reviewer thinks this paper can be accepted for publication after some minor revisions.

Point 1: The optical setup diagram of the digital holographic camera may be shown, if possible.

We added Figure 1 and a new text on lines 101-127 in accordance with the Reviewer’s comments.

Point 2:  What are the special features (advantages, possibly also challenges) using in situ DHC technology for testing? This should be emphasized.

We added new material on lines 101-114 in accordance with the Reviewer’s comments.

Point 3:  Some examples of recorded holograms and reconstructed images may be shown.

We added Figure 4 on lines 220-222 in accordance with the Reviewer’s comments.

Point 4: The algorithm for estimating the plankton number from the holographic camera results is not clear.  

We added a scheme on Fig. 1 and new material on Fig.1 and lines 149-160 in accordance with the Reviewer’s comments.

We are grateful for all the comments. We hope that as a result our work has become much better.

Reviewer 3 Report

In the manuscript "Underwater Holographic Sensor for Plankton Studies in situ, including rhythmic processes", the authors study the rhythmic processes in plankton ecosystems using data from underwater holographic and complementary probe sensors of several physical variables. The proposal was demonstrated in Lake Baikal during the summertime. According to the authors, the proposed study recovered, interpreted, and verified the results of natural measurements of plankton in situ. In this reviewer's opinion, the revised proposal does not lay within the scope of the Sensors Journal since the proposal is not focused on the underwater holographic sensor (which has been published previously in other works by the same authors). The title of the paper, its abstract, Introduction, and conclusions are not adequately related to the main contribution of the paper:  a study of the rhythmic processes in plankton ecosystems. Moreover, several parts of the manuscript lack scientific rigor.

According to the reasons above and following the guidelines for publication in this Journal, I recommend rejecting this paper. The paper, after substantial improvement, could be resubmitted to another Journal. The following concerns must be fully addressed before resubmission:

1.      According to the title, the proposal is related to an underwater holographic sensor with a particular application. Nevertheless, the authors emphasize the plankton monitoring strategy and data analysis throughout the text. I think the proposal's title should be changed, and the paper should be submitted to another Journal. If remaining in Sensors, the authors must considerably improve the presentation of the underwater sensor by employing more scientific rigor concerning the related sensor's technical details, such as the optical setup employed by the sensor, its imaging performance, the numerical procedures involved in the holographic reconstruction of the recorded images, among many other important factors.

2.      References. The references format is inconsistent: They should be unified under one format for clarity. Also, the paper includes excessive self-citations, which must be avoided according to the MDPI guidelines for academic publication. In general, the amount of self-citations used by the authors is in poor taste.

3.      Abstract. The abstract is extremely short and lacks essential information about the proposal. It does not highlight the salient points from the significant manuscript sections; it does not provide proper context and purpose of the work. Additionally, a statement evaluating and indicating any implications of the results is not provided.

4.      Introduction. There is no scientific rigor in this Introduction. The novelty of the proposal is not established nor even presented. There is no context or description of related works. This section lacks background information and fails to put the study into the larger context of the previously published literature. This section does not provide more detail about the subject area and the rationale for the study and does not state the proposal's aims and objectives.

5.      Problem statement section. All section is devoted to justifying the importance of studying plankton in underwater measurements. However, the final paragraph of this section is the only one in which the proposal is described. What is the novelty of this contribution? How does this paper differentiate from others of the same authors' proposals on the same topic (underwater measurements of plankton using a holographic sensor)?

a.      https://www.spiedigitallibrary.org/conference-proceedings-of-spie/11420/114200I/Underwater-holographic-sensors-for-plankton-studies-in-situ/10.1117/12.2557670.short?_ga=2.183835494.1956751417.1655763535-2050886889.1655763535&SSO=1

b.     https://opg.optica.org/ao/abstract.cfm?uri=ao-58-34-g300

c.      https://www.mdpi.com/1424-8220/21/14/4863/htm

d.     Dyomin V. et al. Hardware means for monitoring research of plankton in the habitat: problems, state of the art, and prospects // OCEANS 2019 - Marseille. IEEE, 2019. P. 1–5.

e.      And others by the same authors.

6.      What is the novelty in what concerns the Sensors journal? What are the scientific findings or new proposals? None of these questions is answered correctly in the first two sections of this submitted paper.

7.      Some parts of the paper are obscure and difficult to understand. For example, the sentence "The paper uses a small-size submersible digital holographic camera (miniDHC) to record a digital hologram of …". According to this writing, the paper records a hologram, which has no sense. Language and the overall writing revisions are mandatory. This sentence is only one example, but proper copy-editing will find more.

8.      The sentence "The study [33] describes certain options for using the miniDHC to solve various problems related to particles.". What do the authors mean by problems related to particles?

9.      The sentence "… an be reliably determined using the chosen algorithm." I'm assuming the authors refer to the hologram reconstruction algorithm. What is the chosen algorithm? What is the physics approach? Do the authors implement a discrete version of the angular spectrum propagator to reconstruct the digital holograms? How does it work? A detailed description of the holographic image retrieval method is necessary.

10.  The sentence: "The probe used in this study is marked as the DHC FOCL probe…". What is the design of the probe? If the probe complements the data provided by the underwater sensor, what is the type of data (and sensors included) in this probe? More technical details of the implementation are required.

11.  Related to the previous concern, the sentence: "In addition to the miniDHC, the following sensors measuring the medium parameters are connected to the DHC FOCL probe: depth pressure transducer, temperature sensor, water conductivity sensor, day/night sensor.". So I understand that the underwater sensor has been previously reported. Also, the sensors attached to the probe are conventional. Then, what is the novelty of the proposal?

12.  The sentence "The granules of this noise appear in the reconstructed images and lead to false activation of the recognition system.". Is there an activation system? How does it work?

13.  The sentence: "To minimize (reduce phase change) this location factor, the working (measured) volume of the miniDHC was chosen to be very small – V?? = 20 ml.". What were the technical criteria to determine this working volume? I assume a statistical quantification of the phase changes on the reconstructed images was performed. Is this correct?

14.  The sentence: “Thus, for k = 5, V?? = 20 ml the measured volume was V? = 0.1 l.”. How that this work? If the sensor is not moving (I understand the location of the sensor is stationary), then the "measured volume" is, in fact, the effective measured volume over time (5 shots within one hour). Is this correct? If positive, what is the interval between acquisitions within the complete hour?

15.  The authors say, "It should be noted that this series of measurements was formed in a data processing center located 5 thousand km from the mooring station and the desktop computer.". If a desktop computer receives the data (I assume that by "data" the authors mean images) far from the station, what is the controller system at the station? Is it a Raspberry system or something like that? This part of the work lacks a proper description.

16.  The authors say that a "large Juday net" was used for sampling. I understand this is the ground truth method to compare (validate) the data provided by the proposed underwater sensor. I think the authors should include another optical microscope in the zone to perform this comparison properly from an imaging point of view (which is the most important aspect of the proposal if we follow the paper title).

17.  The underwater holographic sensor seems to provide data for statistical analysis; nevertheless, there is no detailed description of these procedures in the text. Is it the same procedure reported in other papers? If positive, then again, what is the novelty of the work?

18.  Figure 4. What do the authors mean by "holographic images"? Are these amplitude/intensity holographic reconstructions? Phase reconstruction? Refraction index maps?

19.  At the beginning of section 4 (after an unnumbered equation), the authors say, "… number of zooplankton individuals determined from the i-hologram.". How is this determination performed?

20.  The authors mentioned "time-series measurements" transmitted to the processing center. So the numerical procedure to extract information from the holographic images is performed in the station?

21.  The authors state, "Indeed, in the context of the present study the specificity of the analyzed community gives the right to regularize.". So specificity is not required. Why? What is the reasoning behind this apparent arbitrary decision?

Author Response

Response to Reviewer 3 Comments

In the manuscript "Underwater Holographic Sensor for Plankton Studies in situ, including rhythmic processes", the authors study the rhythmic processes in plankton ecosystems using data from underwater holographic and complementary probe sensors of several physical variables. The proposal was demonstrated in Lake Baikal during the summertime. According to the authors, the proposed study recovered, interpreted, and verified the results of natural measurements of plankton in situ. In this reviewer's opinion, the revised proposal does not lay within the scope of the Sensors Journal since the proposal is not focused on the underwater holographic sensor (which has been published previously in other works by the same authors).

The goals of the “Sensors” Journal state that the Journal covers not only the issues of science and technology of sensors, but also their application. We were trying to describe the application of a sensor based on a digital holographic camera that is used to measure the biorhythms of plankton. Apparently, this comment was caused by our incorrect and not really sufficient title of the article. In this version, in accordance with the comments of the Reviewers, we changed the title into “In situ Measurements of Plankton Biorhythms using Submersible Holographic Camera”.

The title of the paper, its abstract, Introduction, and conclusions are not adequately related to the main contribution of the paper: a study of the rhythmic processes in plankton ecosystems. Moreover, several parts of the manuscript lack scientific rigor.

We revised all sections and made the corresponding changes, which are marked in red in the text of the article. The title was also changed. We believe that all the corresponding changes should improve the meanings and conformity of the text.

According to the reasons above and following the guidelines for publication in this Journal, I recommend rejecting this paper. The paper, after substantial improvement, could be resubmitted to another Journal.

In accordance with the comments of the Reviewers, we significantly revised the text of the article (changes are marked in red). We really hope that this will change the impression of the Reviewer.

The following concerns must be fully addressed before resubmission:

Point 1: According to the title, the proposal is related to an underwater holographic sensor with a particular application. Nevertheless, the authors emphasize the plankton monitoring strategy and data analysis throughout the text. I think the proposal's title should be changed, and the paper should be submitted to another Journal. If remaining in Sensors, the authors must considerably improve the presentation of the underwater sensor by employing more scientific rigor concerning the related sensor's technical details, such as the optical setup employed by the sensor, its imaging performance, the numerical procedures involved in the holographic reconstruction of the recorded images, among many other important factors.

We changed the title. Now it sounds like “In situ Measurements of Plankton Biorhythms using Submersible Holographic Camera”. This better fits the meaning of the study and the material presented. We also expanded the description of the holographic plankton sensor, added some technical details. The changes are made on lines 101-160.

Point 2: References. The references format is inconsistent: They should be unified under one format for clarity. Also, the paper includes excessive self-citations, which must be avoided according to the MDPI guidelines for academic publication. In general, the amount of self-citations used by the authors is in poor taste.

We formatted the references and reduced the number of self-citations.

Point 3: Abstract. The abstract is extremely short and lacks essential information about the proposal. It does not highlight the salient points from the significant manuscript sections; it does not provide proper context and purpose of the work. Additionally, a statement evaluating and indicating any implications of the results is not provided.

We revised and expanded the abstract (lines 13-22).

Point 4: Introduction. There is no scientific rigor in this Introduction. The novelty of the proposal is not established nor even presented. There is no context or description of related works. This section lacks background information and fails to put the study into the larger context of the previously published literature. This section does not provide more detail about the subject area and the rationale for the study and does not state the proposal's aims and objectives.

We have revised the introduction. The objectives and tasks are presented in Section 2. The changes are made on lines 28-99.

Point 5:  Problem statement section. All section is devoted to justifying the importance of studying plankton in underwater measurements. However, the final paragraph of this section is the only one in which the proposal is described. What is the novelty of this contribution? How does this paper differentiate from others of the same authors' proposals on the same topic (underwater measurements of plankton using a holographic sensor)?

a. https://www.spiedigitallibrary.org/conference-proceedings-of-spie/11420/114200I/Underwater-holographic-sensors-for-plankton-studies-in-situ/10.1117/12.2557670.short?_ga=2.183835494.1956751417.1655763535-2050886889.1655763535&SSO=1

b. https://opg.optica.org/ao/abstract.cfm?uri=ao-58-34-g300

c. https://www.mdpi.com/1424-8220/21/14/4863/htm

d. Dyomin V. et al. Hardware means for monitoring research of plankton in the habitat: problems, state of the art, and prospects // OCEANS 2019 - Marseille. IEEE, 2019. P. 1–5.

e. And others by the same authors.

We revised the problem statement section. The changes are made on lines 68-99.

Point 6: What is the novelty in what concerns the Sensors journal? What are the scientific findings or new proposals? None of these questions is answered correctly in the first two sections of this submitted paper.

The novelty of the article is the new application of the known solution. The authors never heard about the use of a digital holographic camera to study the biorhythms of plankton in habitat in a real-time mode.

The text of the first two sections is edited and changed. We tried to take into account the questions and comments of the Reviewer.

Point 7: Some parts of the paper are obscure and difficult to understand. For example, the sentence "The paper uses a small-size submersible digital holographic camera (miniDHC) to record a digital hologram of …". According to this writing, the paper records a hologram, which has no sense. Language and the overall writing revisions are mandatory. This sentence is only one example, but proper copy-editing will find more.

We edited the text of the article in accordance with the comments. In particular, this sentence now sounds as follows: “A small-size submersible digital holographic camera (miniDHC) is used in the study to record a digital hologram of the water volume with suspended particles (working volume in Table 1)”.

Point 8: The sentence "The study [33] describes certain options for using the miniDHC to solve various problems related to particles.". What do the authors mean by problems related to particles?

We amended the text with regard to this comment (lines 134-147).

Point 9: The sentence "… an be reliably determined using the chosen algorithm." I'm assuming the authors refer to the hologram reconstruction algorithm. What is the chosen algorithm? What is the physics approach? Do the authors implement a discrete version of the angular spectrum propagator to reconstruct the digital holograms? How does it work? A detailed description of the holographic image retrieval method is necessary.

As we have already specified, the main focus of this article is not the sensor itself, but its use. Therefore, we believe there is no need to discuss the issues of hologram reconstruction here, since it will essentially complicate and overload the core idea of the study. Moreover, judging by the comments, the Reviewer is well aware of our publications in which this algorithm is described.

The above sentence dealt with the algorithm and reliability of classification. All clarifications on this issue are added to the text (pp. 124-160). Now the meaning of this text has become clearer:

The data for each particle of the 2D measured volume image is entered into the final table for the classification analysis. The classification algorithm used in this study is focused on dimensional and morphological parameters of particles [32]. These data are used to calculate the required characteristics of the plankton community depending on the measurement task.

Works [32, 33] show that plankton taxa, which are most important in terms of determining the ecosystem equilibrium, can be reliably determined using the chosen algorithm.

Point 10: The sentence: "The probe used in this study is marked as the DHC FOCL probe…". What is the design of the probe? If the probe complements the data provided by the underwater sensor, what is the type of data (and sensors included) in this probe? More technical details of the implementation are required.

We have added more technical data on the probe in the text (lines 191-195, as well as lines 223-230).

Point 11: Related to the previous concern, the sentence: "In addition to the miniDHC, the following sensors measuring the medium parameters are connected to the DHC FOCL probe: depth pressure transducer, temperature sensor, water conductivity sensor, day/night sensor.". So I understand that the underwater sensor has been previously reported. Also, the sensors attached to the probe are conventional. Then, what is the novelty of the proposal?

The novelty of the article is the new application of the known solution. The authors never heard about the use of a digital holographic camera to study the biorhythms of plankton in habitat in a real-time mode.

As we noted above, the text of the first two sections is edited and changed. We tried to take into account the questions and comments of the Reviewer.

Point 12: The sentence "The granules of this noise appear in the reconstructed images and lead to false activation of the recognition system.". Is there an activation system? How does it work?

This sentence was changed (explanations are given  on lines 201 to line 210).

Point 13: The sentence: "To minimize (reduce phase change) this location factor, the working (measured) volume of the miniDHC was chosen to be very small – V = 20 ml.". What were the technical criteria to determine this working volume? I assume a statistical quantification of the phase changes on the reconstructed images was performed. Is this correct?

We use quality assessments to determine the working volume. The technical criteria are the characteristics of the reconstructed images: the recognition ability along the length of the boundary and the area of the particle image cross-section. The values of these characteristics are determined by the water turbidity in the water area and the length of the working volume. The studies of this issue are presented in our previous works. Since this does not apply to the subject of the article, we did not give a detailed description of these studies to avoid overloading of the article with data.

Point 14: The sentence: “Thus, for k = 5, V = 20 ml the measured volume was V = 0.1 l.”. How that this work? If the sensor is not moving (I understand the location of the sensor is stationary), then the "measured volume" is, in fact, the effective measured volume over time (5 shots within one hour). Is this correct? If positive, what is the interval between acquisitions within the complete hour?

Yes, this is so, the sensor does not move, but plankton moves. The time between adjacent holograms to form a measured (or effective) volume is about 5 seconds (added on lines 213-215). During this time, the plankton will shift by a distance greater than the cross-section of the measured volume, i.e. the scene will be reliably changed from frame to frame.

Point 15: The authors say, "It should be noted that this series of measurements was formed in a data processing center located 5 thousand km from the mooring station and the desktop computer.". If a desktop computer receives the data (I assume that by "data" the authors mean images) far from the station, what is the controller system at the station? Is it a Raspberry system or something like that? This part of the work lacks a proper description.

We made the necessary adjustments and clarified certain points in the text (lines 223-232).

Point 16: The authors say that a "large Juday net" was used for sampling. I understand this is the ground truth method to compare (validate) the data provided by the proposed underwater sensor. I think the authors should include another optical microscope in the zone to perform this comparison properly from an imaging point of view (which is the most important aspect of the proposal if we follow the paper title).

An optical microscope is used here. But since it is located in the laboratory and is intended to consider plankton samples, it is not discussed in the text of this work. This is a standard procedure for processing plankton samples.

Point 17: The underwater holographic sensor seems to provide data for statistical analysis; nevertheless, there is no detailed description of these procedures in the text. Is it the same procedure reported in other papers? If positive, then again, what is the novelty of the work?

Yes, this is the same procedure described in other works, including ours. The novelty discussed in this work is the study of plankton biorhythms in the habitat using a digital holographic camera. See response to item 6 of the Review.

Point 18: Figure 4. What do the authors mean by "holographic images"? Are these amplitude/intensity holographic reconstructions? Phase reconstruction? Refraction index maps?

The holographic image here means the distribution of intensity in the image plane reconstructed from a digital hologram. We added an explanation on lines 124- 160, and added Fig. 1 for more clarity.

Point 19:  At the beginning of section 4 (after an unnumbered equation), the authors say, "… number of zooplankton individuals determined from the i-hologram.". How is this determination performed?

We amended the text to clarify this issue and added a scheme in Fig. 1b.

Point 20: The authors mentioned "time-series measurements" transmitted to the processing center. So the numerical procedure to extract information from the holographic images is performed in the station?

Yes. It is performed in the station. This was amended on lines 223-232.

Point 21: The authors state, "Indeed, in the context of the present study the specificity of the analyzed community gives the right to regularize.". So specificity is not required. Why? What is the reasoning behind this apparent arbitrary decision?

This decision is justified by two points, which are addressed in the article:

- poor  biodiversity of the plankton of Lake Baikal.

- different species appear in the field of view of the device at different times.

This is shown in the relevant section (lines 250- 254).

In our case, the violation of this specificity was caused by the appearance of algae (phytoplankton) in the field of view of the camera, and the regression line described the changes in plankton number that occurred as part of the process. It is clear that these changes occur in other rhythms, for example in infradian rhythms. Since this is the topic of another work, we did not go into these details, thus only limiting ourselves to the regularization procedure.

This is described in the text of the article on lines 278- 287.

We are grateful for all the comments. We hope that as a result our work has become much better.

Round 2

Reviewer 3 Report

Although the authors have made an effort to improve the quality of the manuscript, the novelty statement and other key points are still missing in this proposal. In particular:

- In the introduction and problem statement sections, although improved, it is not clear what's the novelty of the work with respect to other works by the same authors:

https://www.frontiersin.org/articles/10.3389/fmars.2020.00653/full

This comment has not been addressed by the authors.

- Technical details of the sensor are still missing. According to the authors, this is not the main interest of the proposed paper.

- The authors state: "The authors never heard about the use of a digital holographic camera to study the biorhythms of plankton in habitat in a real-time mode." What about their paper: https://www.frontiersin.org/articles/10.3389/fmars.2020.00653/full? I believe the distinction between the current proposal against that paper is not presented. According to the authors, this article is not about the sensor itself but its use. Considering that its use has already been reported in a previous work by the authors, I believe this is merely an incremental paper. I respectfully say that I do not believe the paper meets the requirements for novelty.

- Although I understand that the authors claim that the paper's interest does not rely on the sensor, I believe points 13 and 18 have not been thoroughly addressed. This question derives from more questions: How was the best focal plane determined? Visually? How do the authors process the recorded digital holograms to produce the images shown in Figure 1? These images were used afterward for the statistical analysis. Nevertheless, there is no physical (optical) explanation of the numerical procedure.

- Point 16. According to the author's response, there is no actual validation of the measurements for the particular in-situ device.

- Point 17. As for the concern, "The underwater holographic sensor seems to provide data for statistical analysis; nevertheless, there is no detailed description of these procedures in the text" the authors say, "this is the same procedure described in other works, including ours".
